# Corrosion-Resistant Plug Materials for Geothermal Well Fluid Control

**DOI:** 10.3390/ma15196703

**Published:** 2022-09-27

**Authors:** Bill Lowry, Andrew Louden, Robin Jerman, Tatiana Pyatina

**Affiliations:** 1isol8 Inc., Port Townsend, WA 98368-9305, USA; 2isol8 Ltd., Aberdeen AB12 3SN, UK; 3Brookhaven National Laboratory, Upton, NY 11973-5000, USA

**Keywords:** thermite-based sealers, corrosion resistant well materials, cement alternatives, geothermal wells, CO_2_ resistant well materials, thermal shock, anorthite, mullite, aluminosilicate sealer

## Abstract

Conventional cements and plugs are challenged by corrosion in CO_2_-rich and extreme geothermal environments, due to the hostile chemistry and high temperatures. Thermite-based sealing and well intervention technologies are being applied in the oil and gas industry, combining the energy delivery capability of thermite materials with the sealing characteristics of low melt temperature alloys. The thermite reaction products (ceramics) and the sealing alloys used in these plugs both have very attractive corrosion properties, and their operating envelopes extend into geothermal conditions. Thermite plugs and platforms, without supplemental sealing materials, have been considered for nuclear waste isolation, carbon sequestration, and geothermal applications due to the geochemical stability of the ceramic product and its very high service temperature. This study addresses corrosion resistance of the thermite reaction products. A range of engineered thermite systems which yield thermite reaction products including pure aluminum oxide, feldspar, or aluminosilicate solid solutions (in addition to the iron produced in thermite reactions) was developed. These materials were evaluated for their strong acid resistance (pH 1), carbonate resistance (sodium carbonate) and thermal shock resistance (600 °C heating → cold water quenching repeated three times). Performance of different materials was evaluated based on the changes in mechanical properties, water-fillable porosity, phase changes under stress conditions. The aluminosilicate product exhibited very good corrosion resistance, both from material loss and strength perspectives, while the other products performed with varying degrees of stability. This paper presents the results of the thermite corrosion studies and describes the novel tools being deployed, and under development, to satisfy challenging barrier and intervention applications.

## 1. Introduction

Plug and abandonment (P&A) of underground wells is widespread and has multiple challenges to form permanent seals under a variety of field conditions [1]. Well construction methods vary in the oil and gas, geothermal, sequestration, and nuclear waste applications and pose different challenges to barrier emplacement, with varying performance requirements. An example is the challenge of CO_2_ storage in underground wells [2,3,4,5]. Commonly used Portland cement-based materials are not stable in the environments of super-critical CO_2_ and high CO_2_ concentrations due to calcium reactions with CO_2_ with formation of calcium carbonate followed by the formation of soluble calcium bicarbonate after longer exposures. Dissolution of calcium leaves amorphous silica gel in cement, which results in increased porosity and weakening of the cement matrix. Several recent review papers discussed underground well degradation and laboratory simulations of this process under CO_2_-rich conditions [6,7,8].

Geothermal wells present additional challenges for cements to survive, including strong acids, high dissolved solids concentrations, and thermal shock conditions [9]. New materials have been developed and tested for geothermal wells with varied success [10,11,12,13].

Thermite plugs and platforms have been evaluated for nuclear waste isolation, carbon sequestration, and geothermal applications [14], where they can form high strength, corrosion resistant, structural platforms in wells [15,16]. The previous research tailored the thermite formulations to form ceramic or mineral-like products geochemically compatible with the surrounding rock. These plugs perform structural and flow control functions and are particularly suited to uncased wells where they can fill irregular boreholes and couple to the native rock. As will be shown in this paper, the corrosion resistance, thermal shock resistance, and service temperature of these thermite products far exceed the performance of conventional well cements and show promise in the more extreme downhole environments.

Where higher performance seals are required, such as gas-tight high-pressure seals in plugging and abandonment of metallic tubing, thermite plugs are complemented by other sealing media such as low melt temperature alloys. We are developing and fielding tools combining the high energy delivery capability of thermite with the flowing and sealing characteristics of fusible alloys. Current tool development is addressing tubing P&A applications in relatively modest temperature environments (up to 160 °C) but much higher applications are on the horizon, with HP/HT being the ultimate objective. Thermite plugs, without alloy seal media, have been demonstrated in full scale laboratory tests but have not yet been demonstrated in field applications. 

The Fusion tool architecture is depicted in Figure 1. The modular design supports a range of applications:Thermite-only ceramic plugs using consumable thermite housingsFusible metal plugs emplaced using retrievable thermite heatersFusible metal seals in annuli between tubing and casing

The Fusion tool is emplaced via wireline, slickline, or coiled tubing, and the control system permits autonomous or surface-controlled operation. If needed, a retainer module forms a fixed anchor platform upon which the plug forms (not necessary if the plug is placed above bridge plugs, cement platforms, or fill).

In Figure 2 the formation process for a thermite-only plug is shown. The tool consists of a control module at its top, a consumable thermite module, and a retainer at its base (the retainer is not required if the plug is set on top of an existing platform). The tool is lowered to the setting depth and the reaction initiated at the base of the thermite module. This initiation sets off a sequence of mechanical actions in the tool. First, the retainer anchors are deployed and set to hold and centralize the tool in the borehole. Above that a retainer basket is deployed to capture mobile components of the plug in the subsequent reaction. The thermite reaction progresses from the bottom of the thermite module upward, reacting and fluidizing the thermite as the module sinks into the reaction product pool. In this case, the thermite is formulated to produce a fluid, magma-like product which flows freely to fill the well volume. During the reaction the thermite modules disengage from the control module, allowing the thermite modules to sink into the reaction product, similar to a welding rod feeding into a molten weld pool. The control module is retrieved to the surface and the thermite plug cools and solidifies.

In Figure 3 the integration of thermite heater and fusible metal alloy is depicted. The thermite is housed in a non-consumed housing. When initiated the thermite module remains in place during reaction while metal alloy beads are released from the alloy housing above, surrounding the heater elements. The alloy melts to fill the tubing and at a prescribed time the retrievable parts of the toolstring (control module, alloy housing, and thermite modules) are pulled to the surface.

Thermites are a class of energetic material which combines a combustible metal with an oxide to release large amounts of thermal energy (Figure 4) [17]. A common formulation is the aluminum/ferric oxide system, where aluminum is the combustible metal and ferric oxide (Fe_2_O_3_) the oxidizer, yielding iron and aluminum oxide products, and a large amount of energy upon reaction. The reactants are blended in powder form, compacted in metal housing, and ignited when placed at the desired location in a well. The reaction is designed to progress at a relatively slow pace (fractions of a cm/s. Aluminothermite is insensitive as an energetic material due to its high activation energy, but once started will tend to react to completion (even under water, since it is self-oxidizing). The stoichiometric aluminothermite reaction reaches a peak temperature of approximately 2900 °C. Additives can be combined with the thermite to alter peak temperature and tailor the final product form. High temperature oxides and mineral additives reduce the peak temperature, moderate the reaction rate, and combine with the thermite reaction products to yield other compounds, such as feldspars. The iron will either be distributed throughout the ceramic product matrix or accumulate at the bottom of the plug, in the case of the more fluid product forms. This study evaluates only the ceramic portion of the thermite reaction product, which constitutes typically over 70% of the product volume.

Three ceramic product forms were selected for this corrosion study:Aluminum oxide—the product of the pure thermite reaction, or the aluminothermite formula diluted with aluminum oxide.Aluminosilicate—also called mullite, the product of the aluminothermite reactants diluted with SiO_2_.Anorthite—a feldspar material resulting from addition of specific oxides and minerals to the aluminothermite reactants.

## 2. Materials and Methods

Thermite ceramic product samples were prepared by reacting the ingredients in a cylindrical container, allowing it to cool, and then sectioning it to the sizes required for testing. The basic aluminothermite material was a combination of aluminum powder and iron oxide powder in a stoichiometric ratio (approximately 1:3 by mass of aluminum to iron oxide). In all cases the basic aluminothermite formulation was diluted with up to 35% additional oxides and minerals. For the aluminum oxide product, aluminum oxide formed the diluent (100 mesh alumina (Panadyne Inc., Montgomeryville, PA, USA)). The aluminosilicate product was formed by addition of silica sand (70 mesh quartz sand, Lane Mountain Company, Valley, WA, USA), which combines with the alumina product to form an aluminosilicate solid solution called mullite. Feldspar product is formed by adding silica sand and calcium metasilicate (Seaforth Mineral & Ore Co., Inc., Cleveland, OH, USA), which combine with alumina to form anorthite.

Samples were formed under atmospheric conditions. All thermite ceramic product samples were evaluated “as received” or “before exposure” samples, after autoclaving at 250 °C overnight, and after the exposure/stress tests. The exposure tests included 30 days of strong acid (sulfuric acid pH 1 prepared from concentrated sulfuric acid received from Sigma-Aldrich, St. Louis, MO, USA) at 90 °C, sodium carbonate (20 days in 0.05 M solution and 20 more days in 0.1 M solution at 250 °C, both prepared from sodium carbonate obtained from Sigma-Aldrich St. Louis, MO, USA), three thermal shock cycles (1 cycle: 600 °C heating for 24 h → cold water quenching for 10 min). After the first 20-day exposure to alkaline carbonate conditions, the concentration of sodium carbonate solution was increased to accelerate samples degradation. Reference Portland cement-based samples were prepared by dry-blending class G (Dyckerhoff, North type, Wiesbaden, Germany) Portland cement with 35 wt.% silica flour (both from Schlumberger Inc., Sugar Land, TX, USA). The slurry was mixed at 0.44 water-to-cement ratio and poured into cylindrical molds of 40 mm high and 20 mm in diameter. The OPC-based samples were firstly cured at 85 °C and 99 ± 1% relative humidity overnight (to imitate slurry placement), then autoclaved at 250 °C overnight. 

In the acid exposure tests the volume of acid was 3-times that of the samples and the acid was changed every second day. Electromechanical Instron System Model 5967 (Norwood, MA, USA) was used to obtain samples compressive strength. The Thermite-based samples had cubic shape with the side of about 20 mm. Anorthite samples were too strong to be tested with the available machine, so about 1/3 of each sample was cut off for the tests of mechanical properties. At least 3 samples were used for each measurement.

XRD (40 kV, 40 mA copper anode X-ray tube), Attenuated Total Reflectance-Fourier Transform Infrared Spectroscopy (ATR-FTIR, Perkin Elmer, Waltham, MA, USA), Energy Dispersive Micro X-ray Fluorescence Spectrometer (µEDX, Shimadzu, Columbia, MD, USA) were used for samples characterizations. The results of XRD tests were analyzed using PDF-4/Minerals 2021 database of International Center for Diffraction Data (ICDD). Nikon Eclipse LV 150 3-D microscope was employed to obtain visual images of tested samples before and after the acid exposure. Additionally, JEOL 7600F Scanning Electron Microscope (SEM) (Pleasanton, CA, USA) image analyses coupled with EDX elemental composition survey were done for the typical representative spots on freshly broken samples. Cement and mineral samples were coated with silver or/and gold to decrease the charging effects.

Changes in water fillable porosity of the samples were calculated in the following manner. The excess water was wiped of the surface of autoclaved samples with a paper towel and the samples were weighed. Then, the samples were placed in a vacuum oven for 3 days at 60 °C to eliminate removable free water. The free water content, wt.%, was computed by W_wet_ − W_dry_/W_wet_ × 100, where W_wet_ is the weight of a wiped water-saturated sample after the autoclave and W_dry_ is the weight of the dried sample. After the exposure tests, the samples were immersed in water at ambient temperature for at least 4 days until the weight of the samples became constant. Afterward, the free water was calculated in the same manner as that of the samples before the exposure tests. 

## 3. Results

### 3.1. Composition of the Samples before Exposure Tests and 250 °C Autoclaved Samples

The major elemental and crystalline composition analyses (µEDX and XRD, respectively) of the thermite ceramic samples before the exposure tests are shown in Table 1. The elemental composition is given as an average of 3 different sites. The data show that all the samples had some iron inclusions. Additionally, aluminum oxide samples had very small impurity of calcium and manganese, anorthite samples had inclusions of chlorine, and aluminum-silicate samples of manganese, calcium, and chromium. 

The crystalline phase analysis confirmed that the main component of aluminum oxide samples was corundum; Anorthite samples were mostly anorthite mineral with some inclusions of hercynite (iron-aluminate) and corundum. Aluminum-silicate samples were mostly mullite/sillimanite with some iron-aluminum-silicate and inclusions of crystalline iron oxide. The aluminum-silicate samples will be referred to as “mullite” further on.

The autoclaving of these samples at 250 °C did not change their elemental and crystalline compositions.

### 3.2. Acid Exposure Tests

#### 3.2.1. Changes in Samples Appearances

Photographs of the samples after the acid exposure are shown in Figure 5. Mullite kept its original appearance. Anorthite had some white patina and light erosion of the edges. The acid solution was turbid after anorthite’s exposure suggesting that the patina was formed by sample dissolution/erosion and some precipitate from the acid. The porosity of aluminum oxide sample visually increased and its cohesion decreased. The samples were covered with holes of various diameter and slightly crumbling when handled. Portland cement-based formulation was covered with a white layer, which was scaling off on the ends and partially covering the sample in the middle. The fact that the white layer was denser at the ends of the samples, that were exposed to the acid standing, suggested that it could be a precipitate from the acid solution of cement-acid reaction products. The corrosion layer of the sample was uneven, on average measuring ~1 mm.

Higher magnification of the sample surfaces revealed slight decrease in mullite’s surface crystallinity (Figure 6). The figure shows three surface locations (a–c). It includes sample’s surface photographs and its 3D images (black background) with the numbers indicating the depth of the surface features. The top row shows samples’ appearance before the acid exposure, the bottom row shows appearance of the same site after the acid exposure. (Note, only 3D images before (top, b) and after (bottom, b’) the acid exposure are shown for the second location with the shiny metal inclusion). Despite some signs of decreased sample crystallinity, the depth of the surface features remained either unchanged (Figure 6, sites a, b) or increased only slightly, due to the minor sample erosion (Figure 6, site c). Metal inclusion into the mineral, shown by µEDX to be iron, remained intact after the 30 days of strong acid exposure, which confirmed only very minor erosion of the sample (Figure 6, site b).

Both erosion and precipitation can be seen on anorthite samples after the acid exposure (Figure 7, images a’ and b’). The sample erosion resulted in a clear dark straight-line pattern, while precipitation created white patina on the sample surface. The depth of the surface features slightly decreased after the exposure due to the precipitation of the anorthite-acid reaction products on the surface of the samples. However, the integrity of the sample was intact.

In the case of aluminum oxide both erosion and precipitation of aluminum oxide-acid reaction products on the sample affected the appearance and the depth of surface features of the sample (Figure 8). The crystallinity of the sample decreased, which can be clearly observed on a photomicrograph of a typical surface shown in Figure 8, site a. The surface features smoothed out and their depth slightly decreased as is visible from the 3D images showing decreased depth profile of the surface samples from 1.6 mm to 1.55 mm due to the precipitation of corrosion reaction products. In other places the depth of the sample surface features increased due to the erosion. The depth increase was especially noticeable in the locations of iron presence in non-treated samples. Photographs in Figure 4 show that site b lost the iron inclusion because of the sample erosion around the inclusion during the acid exposure. 

In summary the visual inspection of the samples exposed to a strong acid showed that materials’ acid resistance increased in the following order: aluminum oxide < anorthite < mullite. 

#### 3.2.2. Changes in Weight and Mechanical Properties

Figure 9 shows changes in weight of the samples exposed to the strong acid as a function of the exposure time. Only OPC-based sample gained some weight due to the significant scaling. The weight gain was not linear, some losses were observed between 10 and 20 days of the exposure for this sample. They were likely caused by the loosely attached scale peeling off the samples. By the end of the 30-day exposure the OPC/SiO_2_ samples added on average 1.5% of the weight. 

The rest of the samples underwent surface erosion accompanied by the weight loss (negative numbers of percent weight change in the figure). The losses were most significant for aluminum oxide samples. On average aluminum oxide lost more than 9% of its weight after the 30-day exposure. The 3-D images of the aluminum oxide revealed that part of the weight loss could be associated with the loss of the iron inclusions in the samples. Mullite weight losses were the smallest at less than 2%. Anorthite lost about 6% of its weight.

The EDX analysis of sulfur accumulated at the samples’ surfaces during the acid exposure showed 56% sulfur (expressed as SO_3_) in the middle and 22% sulfur at the ends of the samples for OPC/SiO_2_, 14 ± 5% SO_3_ was detected at the surface of anorthite samples. Mullite and aluminum oxide samples did not have any sulfur accumulation on the surface. The degradation of these samples took place through their erosion. Unlike OPC/SiO_2_ and anorthite, these materials did not have calcium in their composition, which reaction products with sulfuric acid could precipitate on the samples in the form of sulfates.

The high sulfur content of the OPC-based samples at the middle location was clearly associated with the formation of calcium sulfate. The µEDX analysis did not detect any silica there, while the calcium content decreased from 64% CaO before the exposure to 44% CaO after the exposure. Dissolution of calcium through reactions with sulfuric acid was followed by formation of calcium-sulfate layer on the samples’ surfaces. Silica released from calcium-silicate compounds precipitated on the ends of the samples standing during the exposure tests. The composition detected at the ends of the samples expressed as oxides included 11% CaO, 67% SiO_2_, and 22% SO_3_. The susceptibility of OPC-based cement formulation to strong acid attack was expected. 

For the rest of the samples, in agreement with the 3D microscope observations of the acid-exposed samples, the weight changes suggested that samples’ acid resistance increases in the following order: aluminum oxide < anorthite < mullite.

#### 3.2.3. Compressive Strength and Porosity Changes

Figure 10 shows results of compressive strength measurements of the samples before and after the acid exposure. The samples of minerals had high compressive strength with the standard deviations significantly larger than normally obtained for cement samples. The strength of aluminum oxide samples decreased significantly during the acid exposure. The strength of “as received” aluminum oxide was 7200 ± 3500 psi; it persisted after the 250 °C curing at 4580 ± 3000 psi but after the acid exposure it dropped to 1960 ± 570 psi. Despite the high standard deviations, without doubt, the cohesion and strength of the aluminum oxide samples dramatically decreased after the acid exposure. Nevertheless, the residual strength of the sample was almost 2000 psi.

The strength of mullite increased from 4280 ± 1070 psi for the “as received” samples to 5760 ± 3790 psi for the 250 °C-cured sample, and further to 9650 ± 4100 psi for the acid-exposed samples. Notwithstanding the high standard deviations, it can be confidently concluded that acid exposure tests did not compromise mechanical properties of the mullite samples.

The strength of OPC/SiO_2_ and anorthite samples persisted through the acid-exposure tests. Anorthite samples had very high compressive strength of more than 20,000 psi. Precipitation of silica and formation of the dense calcium sulfate layer on the OPC/SiO_2_ sample could be partially responsible for its strength persistence. 

Samples’ water-fillable porosity after the exposure decreased in the following order: OPC (23 ± 0.6%) > Al_2_O_3_ (6.3 ± 0.5%) > anorthite (4 ± 1%) > mullite (0.58 ± 0.07%).

#### 3.2.4. Elemental Composition of Sample Surfaces after the Acid Exposure

Elemental ratios of surface composition for tested materials before and after the acid exposure are shown in Figure 11. Changes in the elemental ratios suggest mechanisms of samples degradation under the acid attack, showing which elements were susceptible to the reactions with the acid and removal from the samples (or precipitation on the surfaces). In the case of mullite that was the most stable sample in the strong acid environment, the weight decrease was mostly due to the loss of iron inclusions (decreased Fe/Si ratio). The loss of aluminum was minor, since the Al/Si ratio change was small. Based on 3D photographs of the samples, these changes were likely due to the mineral surface erosion and not due to the precipitation of silicon on the surface.

In the case of anorthite all elemental ratios to silicon decreased, suggesting that silicon depositions were at the surface of the mineral with other elements been removed by the acid attack. The Al/Ca elemental ratio increased from 3.46 to 9.50 suggesting that calcium was more aggressively removed by the acid than aluminum from the sample.

Iron was lost to the acid attack from the aluminum oxide samples in agreement with the microscope observations. This likely happened because of partial aluminum dissolution in the strong acid and sample erosion causing changes in the cohesion of the samples’ matrix in the acid environment. As showed the 3D images of the aluminum-oxide sample exposed to the acid, the loss of iron was not the result of iron interactions with the acid but rather was caused by the loss of iron inclusions into the degrading sample matrix.

#### 3.2.5. Phase Transitions during the Acid Exposure—FTIR and XRD Analyses

The phase transitions taking place during the acid exposure were studied using FTIR and XRD techniques. Figure 12, Figure 13 and Figure 14 show FTIR spectra of the thermite samples before the exposure (“as received”), 250 °C-autoclaved, and acid-exposed mullite, anorthite, and aluminum oxide samples.

Mullite and aluminum oxide did not undergo any phase changes after 250 °C or acid treatments. The patterns are identical (Figure 12 and Figure 14).

Anorthite’s spectrum did not change after the 250 °C autoclaving. However, acid treatment resulted in the transition of crystalline silica into amorphous (increased peaks of amorphous silica at 1140 and 1085 cm^−1^) through calcium removal (Figure 13). The bottom spectrum is from the gel that was collected from the acid solution after anorthite exposure. Amorphous silica was the major component of the gel. The gel was released into the solution after calcium and aluminum removal from the sample through interactions with the strong acid. Silica gel precipitation on the surface of the anorthite samples contributed to the decreased Ca/Si and Al/Si elemental ratios. This agrees with the elemental composition of anorthite surface after the acid exposure showing increase in silicon content. The gel precipitation caused the composition changes and decrease in the depth of surface features discussed above.

Figure 15, Figure 16 and Figure 17 show XRD patterns of the three tested materials. The pattern of the acid-exposed mullite only slightly differed from the control one (Figure 15). There was increase in peak intensity of iron-bearing sillimanite (sillimulite) and iron oxide. There were no crystalline corrosion products (e.g., gypsite). The visual examination of the acid-exposed samples reported above revealed very small changes in acid-exposed samples. In some locations the depth of surface features increased due to minor erosion. This means that the corrosion products formed during the exposure remained in the acid solution with no crystals precipitating at the samples’ surfaces. For the most part the patterns were identical in agreement with other test results showing great resistance of mullite to the strong acid. 

The XRD patterns of crystalline anorthite changed after the acid exposure (Figure 16). The peaks’ intensities of the calcium-bearing anorthite decreased while crystalline silica peaks increased. There was also an increase in the intensities of the peaks of iron-bearing phases—hercynite and iron oxide. The decrease in calcium content through reactions with acid is consistent with the results of µEDX analysis of the sample surface composition showing decreased Ca/Si ratio. Silica was likely released after calcium removal. It partially remained amorphous (FTIR results) and partially crystalized during the 30-day acid exposure at 90 °C.

In the case of aluminum oxide there was clear decrease in sample’s crystallinity—intensity of aluminum oxide and iron-bearing phase hercynite decreased, nearly disappearing for the later (Figure 17). Peaks of manganese-bearing aluminum oxide phase, galaxite became visible due to the general decrease in the intensity of other peaks caused by decreased crystallinity. The disappearance of the iron-bearing peaks agrees with the 3D- images showing loss of iron inclusions from the sample that partially lost its cohesion during the hot acid treatment. In general, the results of this study confirm the results of other analyses reported above. 

#### 3.2.6. Morphological Changes—Scanning Electron Microscope and Energy Dispersive X-ray Analyses of Freshly Broken Surfaces

Figure 18 shows freshly broken surfaces of mullite samples before and after the acid exposure along with the elemental composition of selected representative sites. The composition and morphology of the samples did not change during the 250 °C-autoclaving or 30-day 90 °C strong acid exposure. This observation is consistent with the results obtained with other analytical techniques. 

Figure 19 shows photomicrographs and elemental composition of “as received” (before exposure) and 250 °C-autoclaved anorthite sample. Figure 20 shows representative sites of the sample after the 30-day acid exposure. Sample autoclaving at 250 °C did not change its composition and morphology. The calcium content was around 20%, while iron content was below 10%. After the acid exposure the striking difference was the decrease in calcium content of the sample—from some locations calcium disappeared completely, in others its content varied between 2 and 14%. Iron inclusions appeared to be etched by the acid but still present in the sample matrix (bottom right photomicrograph). The morphology of the sample in general did not change drastically. The site without calcium looked crumbled, but for the most part the matrix’s cohesion persisted through the acid tests even in the parts with low calcium content (bottom left photomicrograph).

Figure 21 shows aluminum oxide samples before and after the exposure. The top middle photomicrograph shows image of iron inclusion in the aluminum oxide matrix. (Note that carbon in the measured elemental composition may be coming from the carbon tape used to attach the samples to the holder.) The matrix of the sample before acid exposure was uniform for both “as received” (before exposure) and 250 °C-exposed samples. 

The samples morphology visibly changed after the acid exposure (bottom photomicrographs). The cohesion of the matrix decreased, separate partially crumbled pieces were clearly visible, no iron inclusions could be detected. The matrix noticeably eroded during the exposure tests. 

For all the tested samples erosion was the main mechanism of acid degradation since no sulfur-containing products were detected in mullite and aluminum-oxide samples and small sulfur precipitation was measured on anorthite samples. The results of morphological-EDX study agreed with the results of other sample evaluation techniques showing increase in acid resistance in the raw: aluminum oxide < anorthite < mullite.

#### 3.2.7. General Notes on Acid Exposure

Among the tested samples OPC/silica and aluminum oxide were strongly affected by the exposure to the sulfuric acid (pH 1, 90 °C, 30 days). The first one underwent scaling corrosion and the later one—acid erosion.

Aluminum oxide sample’s compressive strength decreased by 43% due to the decreased sample crystallinity and cohesion during the acid exposure.

Mullite was very resistant to the acid—its strength persisted, only minor erosion, and insignificant composition and morphological changes were observed for this material.

Anorthite experienced some calcium losses through erosion and precipitation of mostly silica and some calcium-sulfate reaction products on the surface. The sample also likely lost some aluminum, which resulted in silicon release and formation of silica gel and crystalline silica in its composition. The changes were mostly limited to the surface of the sample and did not affect sample’s mechanical properties, which persisted through the exposure tests at nearly 24,000 psi. For the most part, its morphology stayed unchanged. After the acid exposure tests the samples water-fillable porosity decreased in the following order: OPC/SiO_2_ > aluminum oxide > anorthite > mullite.

### 3.3. Alkaline Carbonation at 250 °C and Thermal Shock Tests (3 Cycles of 600 °C Heat → 10 min Cold Water Quenching)

#### 3.3.1. Changes in Samples Appearances

All samples were covered with some white deposition after the alkaline carbonation tests. This deposition was partially an artifact of the test setup due to some dissolution of the glassware used as samples container and silica precipitation on the samples. Partially it was related to the deposition of carbonates on samples surfaces. Because of the possible artifact of glassware dissolution, no further evaluation of the samples’ surfaces was performed. There were no dramatic changes in the samples’ appearance after the 3 cycles of thermal shock tests.

#### 3.3.2. Changes in Mechanical Properties

Figure 22 shows changes in compressive strength of the samples subjected to 40 days of alkaline carbonation. Only anorthite samples lost strength during these tests. The strength of the samples decreesed sed by 44%. Nevertheless, even after the strength decrease the final strength of the anorthite samples was the highest among the tested materials at 12,740 ± 1760 psi.

The strength of mullite samples increased by 117%, and that of aluminum oxide samples increased by 59%. 

The three cycles of thermal shock nearly tripled the strength of mullite samples (it should be noted though that the standard deviation on these samples was very high), while the strength of aluminum oxide persisted within the experimental error. The strength of anorthite samples decreased by 38%, with the remaining strength of anorthite being still very high at 14,100 psi (Figure 23).

These data suggest that the materials resistance to carbonation and thermal shock increased in the following order: anorthite < aluminum oxide < mullite. Mullite’s resistance to both environments was outstanding, the material increased its strength while been stressed.

The strength of all the tested materials was very high both before and after the imposed stresses. This parameter alone would suggest that all tested samples were very resistant. 

To get a more complete performance evaluation it was of interest to compare changes in water-fillable porosity under the stress conditions. The data on water-fillable porosity are shown in Figure 24. Porosity of OPC/SiO_2_ high-temperature formulation is given along with the tested materials for comparison. The porosity of the tested materials was significantly lower than that of OPC/SiO_2_ control sample. Mullite porosity was three orders of magnitude lower than that of Portland cement-based formulation, anorthite—2 orders of magnitude lower, aluminum oxide’s porosity was an order of magnitude lower than that of OPC/SiO_2_ control. After the acid exposure the porosity of all the samples increased but stayed at least an order of magnitude lower than that of the OPC/silica blend. The carbonation decreased porosity of OPC/SiO_2_ and anorthite samples, did not change porosity of the aluminum oxide one and slightly increased porosity of mullite. Even after the slight increase in the porosity it was the lowest for the mullite sample. (It should be noted that partial dissolution of glassware used as samples’ container in the autoclave and silica deposition on the samples could be responsible for porosity changes under the alkaline carbonate conditions).

The thermal shock conditions increased porosity of all the samples. Only aluminum oxide porosity increase was larger after the acid exposure than after the thermal shock. For the rest of the samples thermal shock resulted in the largest porosity increase. 

However, in general, porosity of all the tested materials remained significantly lower than that of OPC/SiO_2_ formulation after all the stress conditions. Under all environments it was the lowest for mullite. 

#### 3.3.3. Phase Transitions during the Alkaline Carbonation and Thermal Shock—FTIR and XRD Analyses

Figure 25 shows FTIR spectra of tested materials after 40 days in alkaline carbonate solution at 250 °C and 3 cycles of thermal shock (one cycle: 600 °C heat, overnight → cold water quenching for 10 min). For the most part the spectra of mullite, aluminum oxide, and anorthite did not change after these stress conditions. There was minor carbonation of anorthite in alkaline carbonate tests (small peaks at 1462, 1422, and 876 cm^−1^), while the thermal shock did not produce any carbonates detectable by FTIR. OPC-based formulation showed clear carbonation peaks after both carbonatation and thermal shock tests. Previous study demonstrated carbonation of OPC/SiO_2_ system in TS tests, followed by the cement degradation under the heat conditions [17].

Among the tested samples, only anorthite showed clear changes in its crystalline composition after the alkaline carbonate exposure tests. Figure 26 compares XRD patterns of anorthite samples after the autoclaving at 250 °C, acid-, alkaline carbonate-, and thermal shock tests. The patterns of acid-exposed samples had increased silica peaks as was described above. Those of alkaline carbonate-exposed samples showed additional peaks of sodium-aluminum silicates and aragonite due to the anorthite dissolution and calcium reactions with the carbonate ions from the solution. Sodium from alkaline carbonate solution reacted with aluminum silicate from calcium-depleted anorthite to form new crystalline phases.

In the case of thermal shock tests, the pattern did not change significantly. In both cases all the peaks of the originally present anorthite and hercynite were clearly visible. These results agreed with the FTIR analysis data.

XRD patterns of aluminum oxide are shown in Figure 27. Unlike in the case of the acid exposure where the crystallinity of the aluminum oxide was compromised, so that its peaks significantly diminished, there were no changes in the XRD patterns of the samples exposed to alkaline carbonation and thermal shocks. Crystalline carbonated species were not detected in the exposed samples. All the major peaks remained unchanged confirming stability of this material under these stress conditions. 

Finally, XRD patterns of mullite samples exposed to various stress conditions did not show any changes (Figure 28), suggesting great stability of this material in all tested environments. 

#### 3.3.4. Morphological Changes—Scanning Electron Microscope and Energy Dispersive Elemental X-ray Analyses on Broken Sample Surfaces

Figure 29 shows morphology of typical sites of mullite samples after the alkaline carbonate exposure along with the elemental composition. The appearance of the samples did not change. There were some scattered depositions of small (about 10 microns) carbon-rich particles (spectrum 2) but for the most part the elemental composition of mullite did not differ from that before the exposure.

For the anorthite samples neither sodium, no carbonate was detected in the cracked samples (Figure 30). The sodium aluminum silicates suggested by XRD, and some calcium carbonate detected by both FTIR and XRD analyses were likely present only in the surface layers of the sample. These was true for pure anorthite sites (spectrum 1), aluminum inclusions (spectra 2 and 5), and iron-rich locations (spectrum 3). Somewhat decreased calcium content compared to the original sample was measured at the tested locations (spectra 1 and 4). This observation agrees with calcium removal through reactions with sodium carbonate from the solution. 

There were no changes in appearance or composition of the aluminum oxide sample after the alkaline carbonation (Figure 31).

Figure 32, Figure 33 and Figure 34 show photomicrographs of mullite, anorthite, and aluminum oxide samples after three cycles of thermal shock tests. Mullite matrix morphology was not strongly affected by these tests. The major difference with the morphology of the original samples was the presence of iron-rich round particles in the otherwise uniform matrix (Figure 32, site 2). It is possible that formation of these particles resulted in increased matrix permeability after the tests (2.12% vs. 0.035% for the control).

Anorthite’s photomicrograph after the thermal shock tests is shown in Figure 33. The two sites correspond to the anorthite itself (1) and aluminum inclusion into the matrix (2). Although the matrix did not significantly change its appearance after the tests some voids are visible in both sample features. The porosity of the sample increased after the tests to 12% (control porosity was 0.4%).

Figure 34 shows morphology and elemental composition of aluminum oxide samples after the thermal shock tests. There was no disintegration of the samples’ matrix observed during the acid exposure. Small-featured inclusions (less than 10 microns), that were not detected before the thermal shock tests, were present in the uniform sample matrix (site 2). The increase in the samples’ porosity after the tests was the smallest among the tested materials, with the final water-fillable porosity value of 4.2% vs. 2.2% of the control sample.

#### 3.3.5. General Notes on Alkaline Carbonation and Thermal Shock Test Results

Evaluation of mullite, anorthite, and aluminum-oxide samples exposed to strong alkaline carbonate solution (0.05 M solution for 20 days followed by 0.1 M solution for 20 more days, 250 °C) and to three cycles of thermal shock (one cycle: 600 °C heat overnight → cold water quenching for 10 min) did not show any significant phase or morphological transitions in these materials. The alkaline carbonation caused some surface changes, mostly in anorthite samples through calcium carbonation and formation of sodium aluminum silicates after calcium removal. However, these changes seemed to be limited to the samples’ surfaces. Carbonation products were not detected in mullite and aluminum oxide samples. The compressive strength of mullite samples more than doubled and that of aluminum oxide samples persisted after the prolonged alkaline carbonate exposure tests. The strength of anorthite samples decreased, while remaining very high (above 12,000 psi). Despite their high strength, samples’ porosity increased for all tested materials. Nevertheless, it remained about 2.5 times lower for aluminum oxide, 36 times lower for anorthite, and about 40 times lower for mullite than the porosity of OPC/SiO_2_ high-temperature formulation. In general, all the tested samples showed an excellent resistance to the carbonation.

Thermal shock stress caused significant decrease in strength of anorthite samples. The residual strength of these samples remained more than 14,000 psi. The porosity of tested materials was more sensitive to thermal shock stress than to the alkaline carbonate exposure tests. The effect of thermal shock on the porosity of the anorthite and mullite samples was more important than on aluminum oxide samples. Since the initial porosity of the samples was very low the porosity detected after the thermal shock was still below that of OPC/SiO_2_ samples. It decreased in the following order OPC/SiO_2_ (25%) > anorthite (12%) > aluminum oxide (4%) > mullite (2%). Increase in the porosity of aluminum oxide and mullite samples was not accompanied by the strength decrease.

Based on the evaluated parameters, the resistivity of the tested materials to the thermal shock improved in the following order anorthite < aluminum oxide < mullite.

## 4. Discussion

The tested materials, mullite, anorthite, and aluminum oxide, provided by Iso8l, differed significantly from commonly used high-temperature OPC-based cement formulations. They possessed exceptionally high compressive strength. The strength of anorthite exceeded 20,000 psi and the strengths of aluminum oxide and mullite samples were between about 4500 and 6000 psi. Additionally, their water-fillable porosity was between an order of magnitude (aluminum oxide) to 3-orders of magnitude (mullite) below that of OPC/SiO_2_ high-temperature formulation. These characteristics of the tested materials affected their performance in geothermal stress environments. Even when their property partially deteriorated under the stress conditions the remaining characteristics of these materials were far superior to those of currently used Portland cement formulations.

The chemistries of the main constituents of the samples proved to be advantageous for the tested stress conditions. Previous studies showed improved stability of mullite and anorthite’s polymorph dmisteinbergite, in strong acid-, alkaline carbonate-, and thermal shock environments [18,19,20]. Aluminum is known to have a lower acid sensitivity than calcium in cementitious composites [21]. The current work further confirmed these earlier fundings.

The strong acid attack (sulfuric acid, pH = 1, 90 °C, 30 days exposure) eroded matrix of aluminum oxide samples. The erosion of aluminum oxide matrix resulted in nearly 4 times lower strength of the exposed samples. Loss of iron inclusions from the matrix of aluminum oxide was part of the sample’s disintegration process. Although the first target of the strong acid attack is calcium, at pH levels below about 3–4 and elevated temperatures aluminum oxide will undergo dissolution with the release of aluminum ions. This process compromised integrity of aluminum-oxide samples in the strong acid environment. 

On the other hand, aluminum-oxide samples performed significantly better in alkaline carbonate environment and under the thermal shock stress. The strength of the samples persisted, there was no matrix erosion or samples carbonation, the porosity did not increase during the carbonation and its increase was 1.5 times smaller during the thermal shock tests than during the acid exposure. It can be argued that aluminum-oxide type material is well adapted for most geothermal stress environments except for highly acidic wells.

Mullite was nearly indestructible in all test environments. Although its water fillable porosity increased in all exposure tests (most noticeably in thermal shock tests) the samples’ integrity, high strength and very low final porosity persisted. Most surprisingly, the strength of the samples increased after all exposure tests. The increase in the sample’s porosity was not accompanied by the strength decrease. There were no morphological changes or phase transitions during the exposures. The changes of porosity could be possibly attributed to the changes in the morphology of iron inclusions into the samples of this aluminum-silicate. The aluminum-silicate chemistry of mullite/sillimanite was the most resistant to all the stress conditions mimicking natural geothermal systems.

Anorthite’s stability in the strong acid was intermediate between that of aluminum oxide and mullite. Calcium was the first to be removed from the samples’ matrix. Aluminum followed, and silicon remained in the precipitate on the sample surface and partially crystalize after longer high-temperature exposure times. It’s interesting to note that in the case of anorthite, the pattern of straight lines appeared on the surface of the acid-exposed samples (Figure 7b’). SEM images of anorthite revealed that aluminum-rich inclusions (likely corundum) formed straight-line pattern in anorthite samples (Figure 33, site 2). These observations confirm the fact that the erosion of anorthite samples in the environment of strong acid took place not only through calcium dissolution but also through aluminum removal, which produced straight-line etchings on the samples’ surfaces. The data showed that hercynite (iron aluminum oxide) was more resistant to the acid attack and persisted after the 30-day exposure, while the presence of calcium-bearing anorthite decreased. Nevertheless, the strength of anorthite samples persisted through the acid exposure tests, and its porosity, although increased, was still nearly 6 time below that of control OPC/SiO_2_ samples (4% for anorthite, vs. 23% for OPC/SiO_2_). Matrix cohesion was not affected by the exposure tests, the morphology of the material did not change. This suggests that although calcium in anorthite was susceptible to the acid attack, its partial loss was not detrimental for the integrity of the material. As for other tested materials there was no direct correlation between the strength of the samples and their porosity for anorthite after the acid exposure. In carbonate environment porosity of the sample decreased, which could be an artifact of silica precipitation over the sample’s surface, but the strength decreased as well. The carbonation effect on anorthite was likely connected to the calcium removal as calcium bicarbonate and formation of alkaline aluminum-silicate crystalline products. The thermal shock tests were the most difficult environment for anorthite. They caused both increase in porosity and decrease in its strength. The thermal shock did not change elemental composition of the sample, but morphological analysis showed formation of some voids in the matrix of the material. It should be noted that the strength of the anorthite samples was too high to test the same samples’ geometries as for aluminum oxide and mullite samples. Smaller samples were cut out of the original cubic ones. These smaller samples were about 1/3 of the volume of the samples for the other 2 materials. The decreased size could affect the consistency of compressive strength evaluations in the case of anorthite. In general, the results were accepted because of the standard deviations being on the same order as for the rest of the samples.

## 5. Conclusions

This study evaluated performance of three materials, with the major composition of mullite, anorthite, and aluminum oxide under simulated stress conditions of geothermal wells, including strong acid (sulfuric acid pH 1, 90 °C, 30 days), alkaline carbonate (0.05 M Na_2_CO_3_ for 20 days followed by 0.1 M Na_2_CO_3_ for 20 more days at 250 °C), and 3 cycles of thermal shock (one cycle: 600 °C heat overnight → cold water quenching for 10 min). 

The materials were exposed to the stress conditions after the initial autoclaving overnight at 250 °C. All three materials showed very low initial water-fillable porosity, which decreased in the following order: aluminum oxide (2%) > anorthite (0.4%) > mullite (0.03%). For comparison the water-fillable porosity of high-temperature OPC/SiO_2_ samples was 22%. All samples also had a very high compressive strength that increased in the following order: aluminum oxide (~5000 psi) < mullite (~6000 psi) << anorthite (~23,000 psi). The strength of the high-temperature OPC/SiO_2_ after 250 °C autoclaving was on the order of 3000 psi. 

It was demonstrated that materials’ resistance to a strong acid (samples porosity after the acid attack and residual compressive strength) improved in the following order aluminum oxide < anorthite < mullite. The strength of mullite samples after the acid exposure, reached nearly 10,000 psi. Aluminum oxide underwent matrix erosion and significant strength decrease during the acid exposure tests (residual strength ~2000 psi). The strength of anorthite samples did not change; however, their porosity increased to 4%. Samples erosion was mostly due to the calcium removal through the reactions with the acid.

The thermal shock resistance of the tested samples improved in the order anorthite < aluminum oxide < mullite. Porosity of all samples increased during the thermal shock tests, with the largest increase observed for anorthite samples (12% finial porosity). Anorthite samples also experienced noticeable strength decrease in these tests (the residual strength ~14,000 psi). Nevertheless, all three materials outperformed OPC/SiO_2_ formulation in terms of the final strength and porosity.

The alkaline carbonation tests caused partial calcium removal from anorthite samples with formation of alkaline aluminum-silicates. The porosity remained low for all the tested samples after the alkaline carbonate exposure. The strength of anorthite samples decreased, remaining very high (above 12,000 psi), that of aluminum oxide did not change, and the strength of mullite samples increased (~12,000 psi).

In general, the tested materials performed exceptionally well under the simulated geothermal stress conditions, except aluminum oxide susceptibility to the strong acid environment. The tested samples outperformed the commonly used OPC/SiO_2_ formulation in terms of the persisting high strength and low porosity.

## Figures and Tables

**Figure 1 materials-15-06703-f001:**
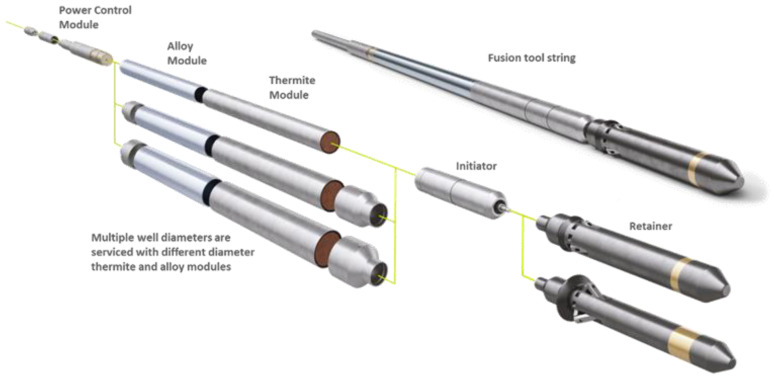
Fusion tool string showing modular design which addresses different well diameters for fusible metal alloy emplacements. For thermite-only plugs, the alloy modules are not included.

**Figure 2 materials-15-06703-f002:**
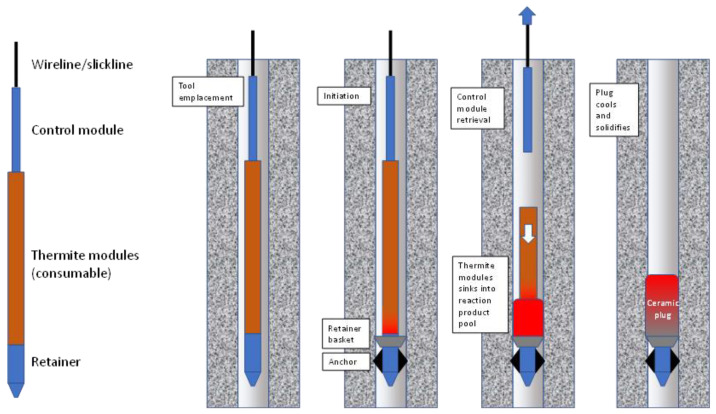
Thermite-only plug emplacement process, leaving the ceramic product in place.

**Figure 3 materials-15-06703-f003:**
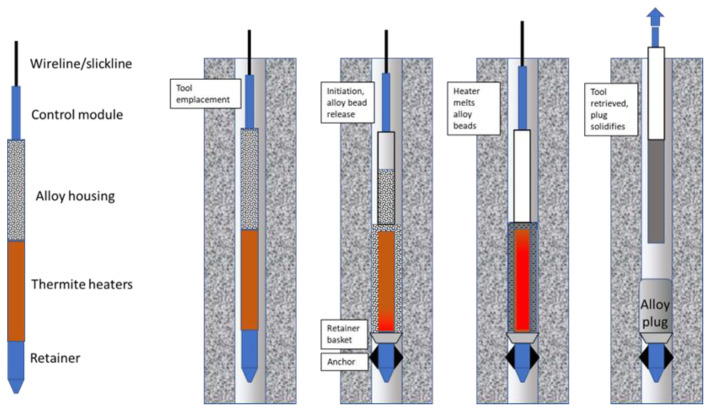
Fusible metal alloy emplacement, where thermite acts only as a removable heater.

**Figure 4 materials-15-06703-f004:**
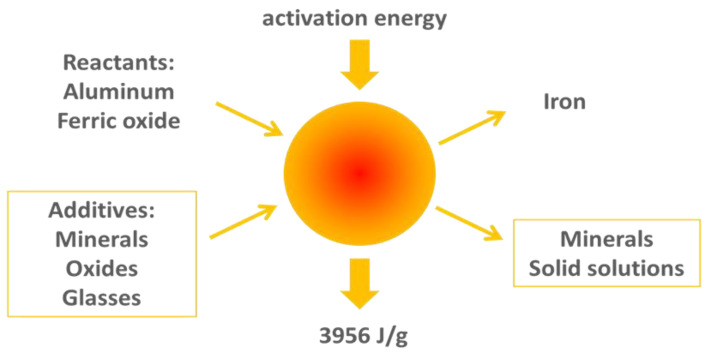
Aluminothermite reactants, products, and energy release.

**Figure 5 materials-15-06703-f005:**
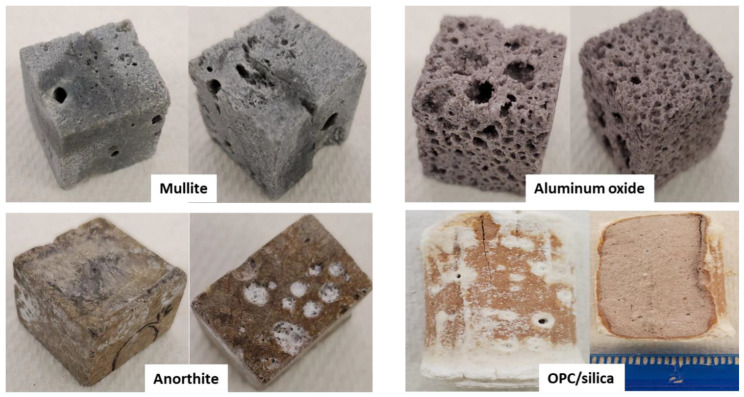
Appearance of samples after the 30-day, pH 1 sulfuric acid exposure.

**Figure 6 materials-15-06703-f006:**
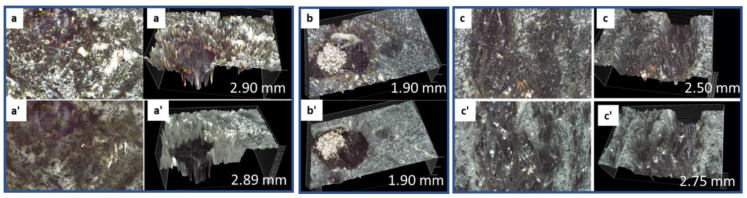
3D images of **mullite** before (top row: (**a**–**c**)) and after (bottom row: (**a’**–**c’**)) the acid exposure for three different sites. The numbers indicate the depth of the sample features on 3D images.

**Figure 7 materials-15-06703-f007:**
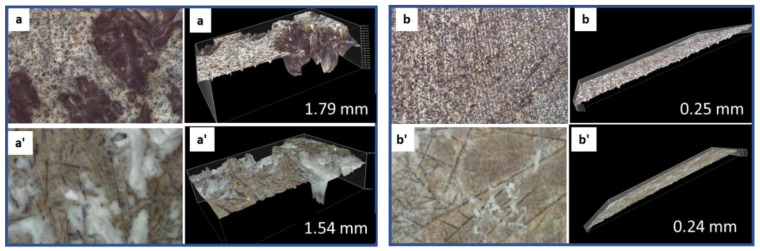
3D images of **anorthite** before (top row: (**a**,**b**)) and after (bottom row: (**a’**,**b’**)) the acid exposure for two different sites. The numbers indicate the depth of the sample features on 3D images.

**Figure 8 materials-15-06703-f008:**
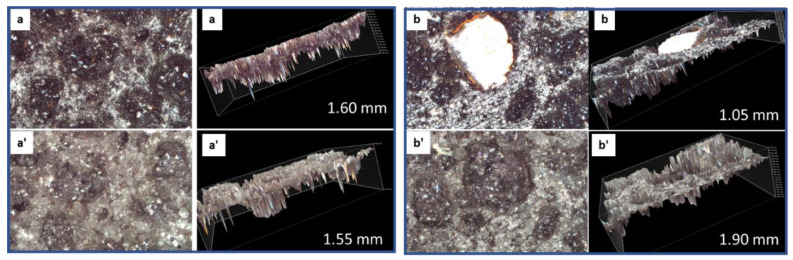
3D images of **aluminum oxide** before (top row: (**a**,**b**)) and after (bottom row: (**a’**,**b’**)) the acid exposure for two different sites. The numbers indicate the depth of the sample features on 3D images.

**Figure 9 materials-15-06703-f009:**
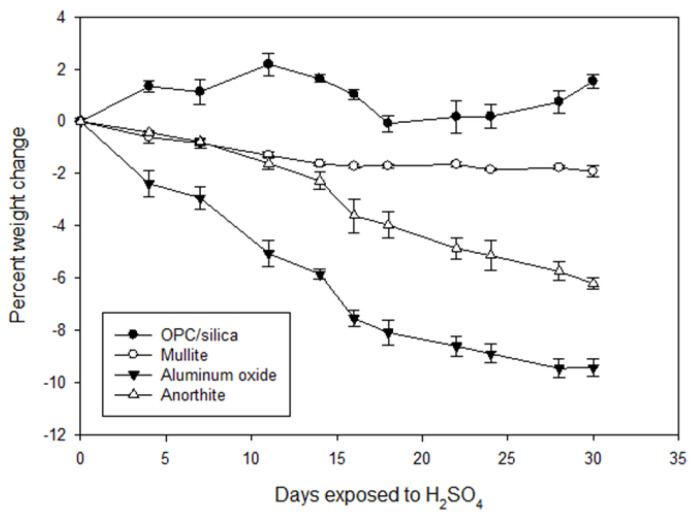
Percent of samples’ weight change in the acid exposure tests (sulfuric acid pH 1, 90 °C) for OPC/SiO_2_, mullite, anorthite, and aluminum silicate samples at different exposure times. The negative numbers signify weight losses and the positive numbers—weight gains.

**Figure 10 materials-15-06703-f010:**
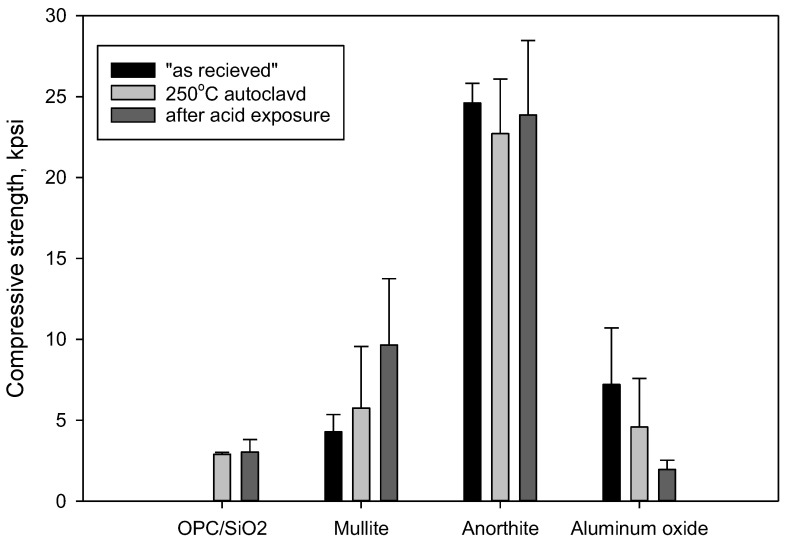
Compressive strength of “as received”, 250 °C autoclaved and acid exposed samples.

**Figure 11 materials-15-06703-f011:**
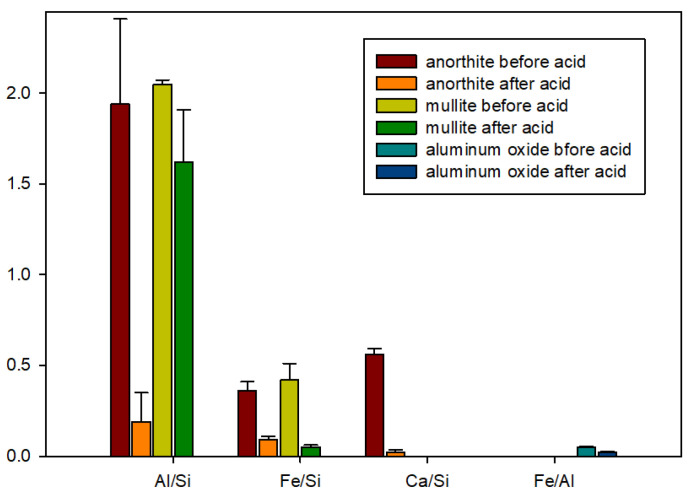
Elemental ratios of surface composition for the tested materials before and after the acid exposure (µEDX measurements).

**Figure 12 materials-15-06703-f012:**
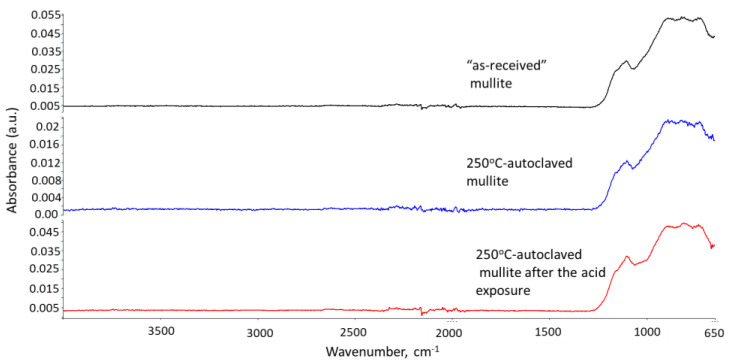
FTIR spectra of **mullite** samples from top to bottom: “as received” (before exposure) mullite, 250 °C-autoclaved mullite and mullite exposed to the strong acid.

**Figure 13 materials-15-06703-f013:**
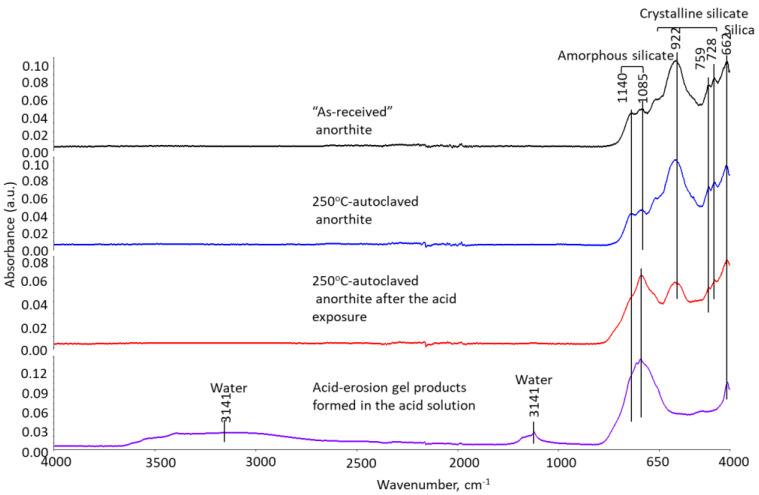
FTIR spectra of **anorthite** samples from top to bottom: “as received” (before exposure) anorthite, 250 °C-autoclaved anorthite and anorthite exposed to the strong acid.

**Figure 14 materials-15-06703-f014:**
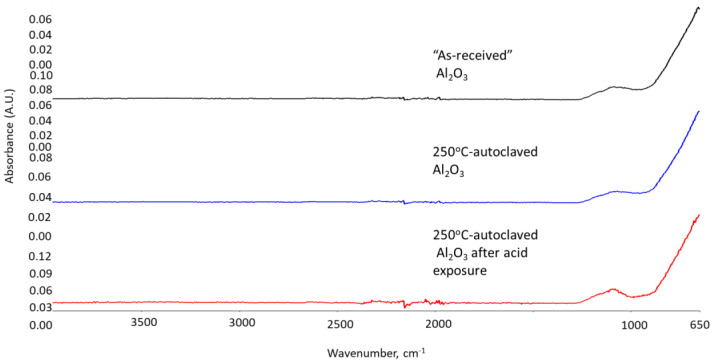
FTIR spectra of **aluminum oxide** samples from top to bottom: “as received” (before exposure) aluminum oxide, 250 °C-autoclaved aluminum oxide and aluminum oxide exposed to the strong acid.

**Figure 15 materials-15-06703-f015:**
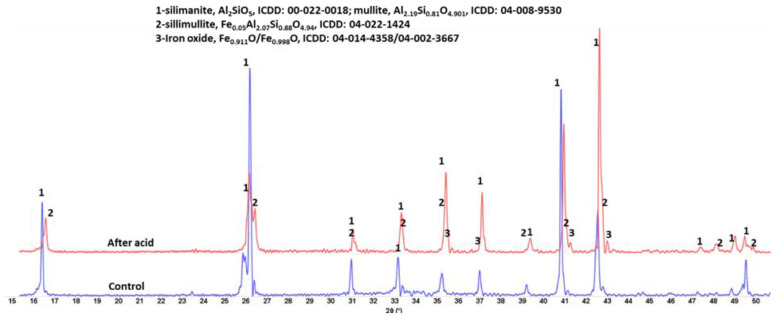
XRD patterns of the control (250 °C autoclaved) and acid-exposed **mullite** samples.

**Figure 16 materials-15-06703-f016:**
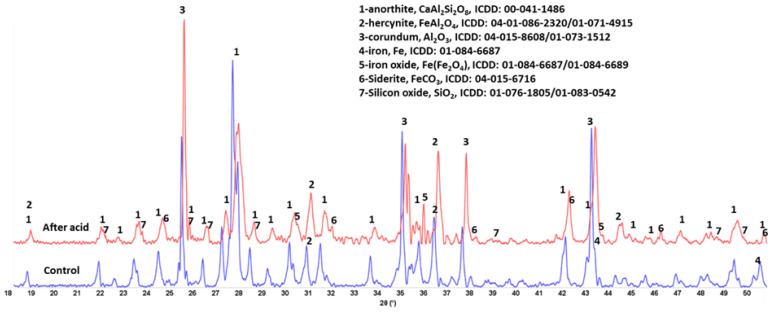
XRD patterns of the control (250 °C autoclaved) and acid-exposed **anorthite** samples.

**Figure 17 materials-15-06703-f017:**
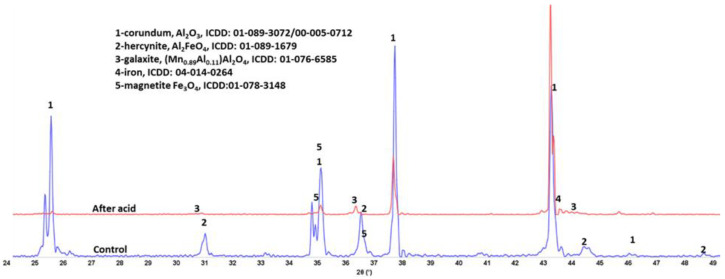
XRD patterns of the control (250 °C autoclaved) and acid-exposed **aluminum oxide** samples.

**Figure 18 materials-15-06703-f018:**
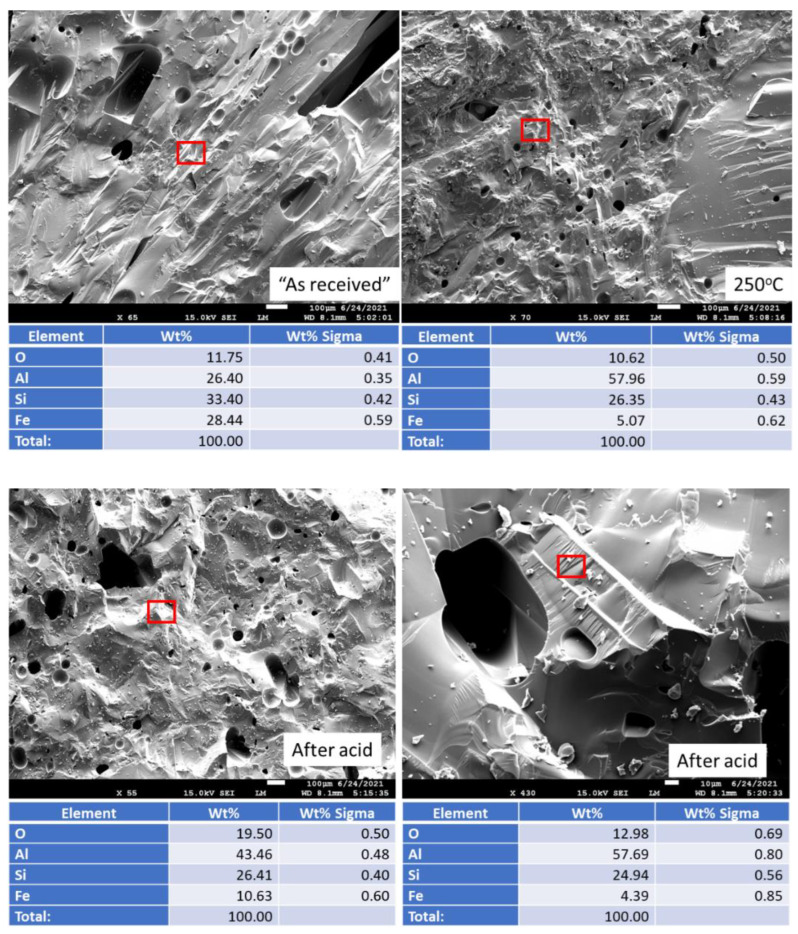
Photomicrographs of **mullite** samples with the elemental composition of typical sites shown by the red boxes.

**Figure 19 materials-15-06703-f019:**
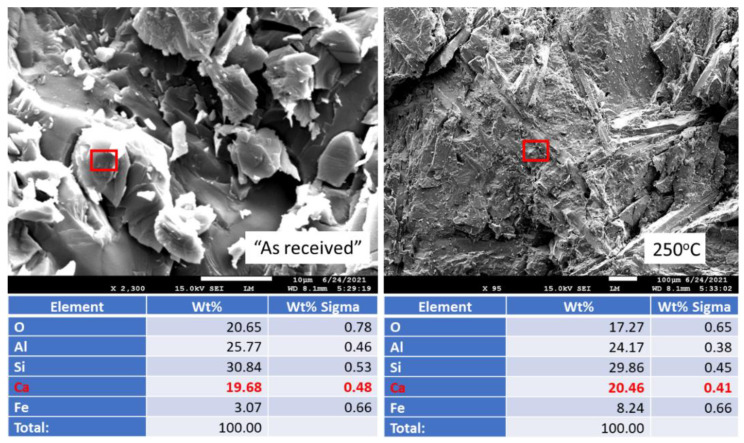
Photomicrographs of “as received” (before exposure) and 250 °C-autoclaved **anorthite** samples with the elemental compositions of typical sites shown by the red boxes.

**Figure 20 materials-15-06703-f020:**
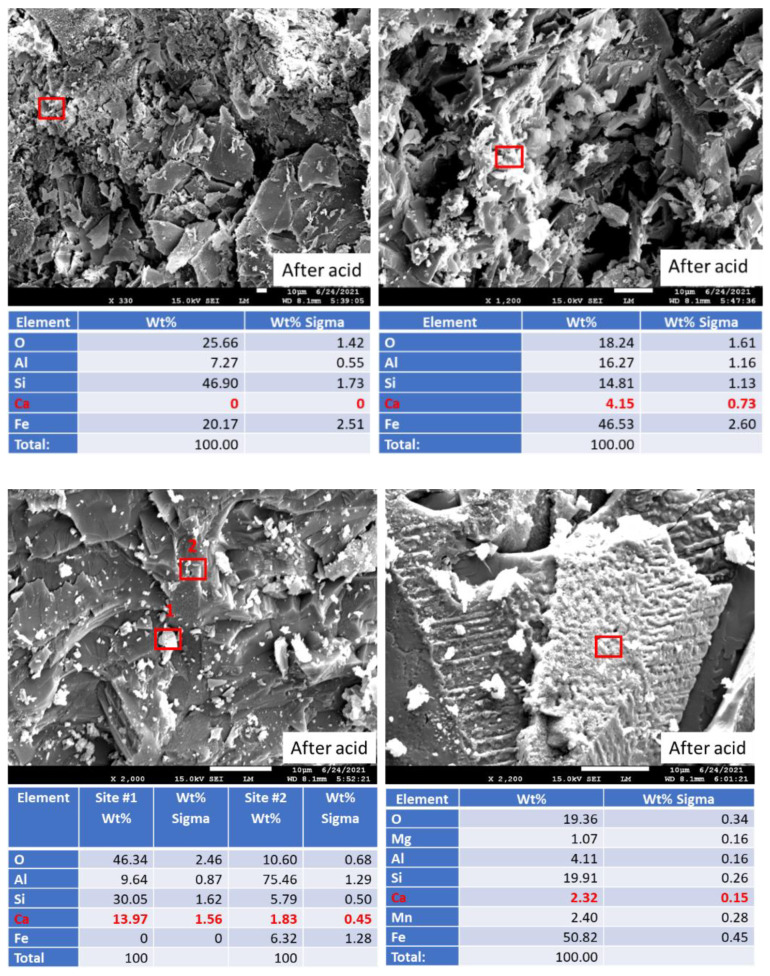
Photomicrographs of acid-exposed **anorthite** samples with the elemental compositions of typical sites shown by the red boxes.

**Figure 21 materials-15-06703-f021:**
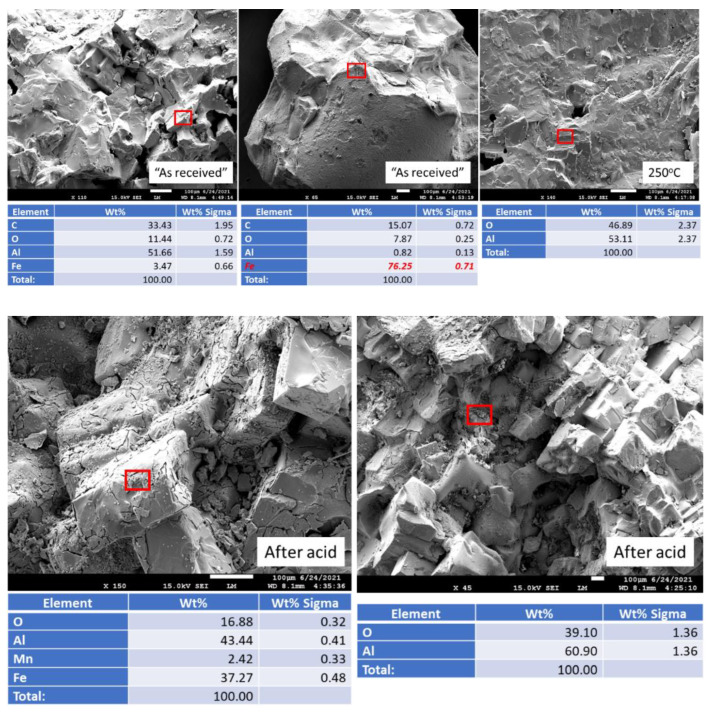
Photomicrographs of **aluminum oxide** samples with the elemental compositions of typical sites shown by the red boxes.

**Figure 22 materials-15-06703-f022:**
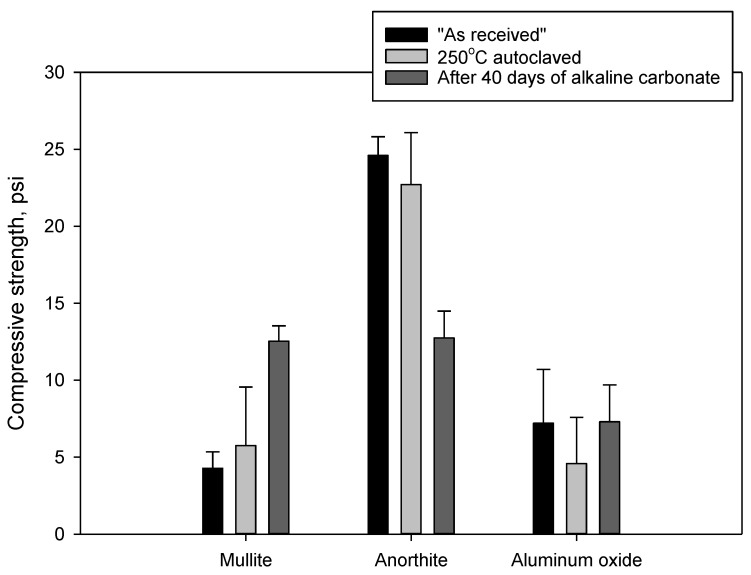
Compressive strength of “as received” (before exposure), 250 °C autoclaved and samples exposed to **alkaline carbonate** for 40 days at 250 °C.

**Figure 23 materials-15-06703-f023:**
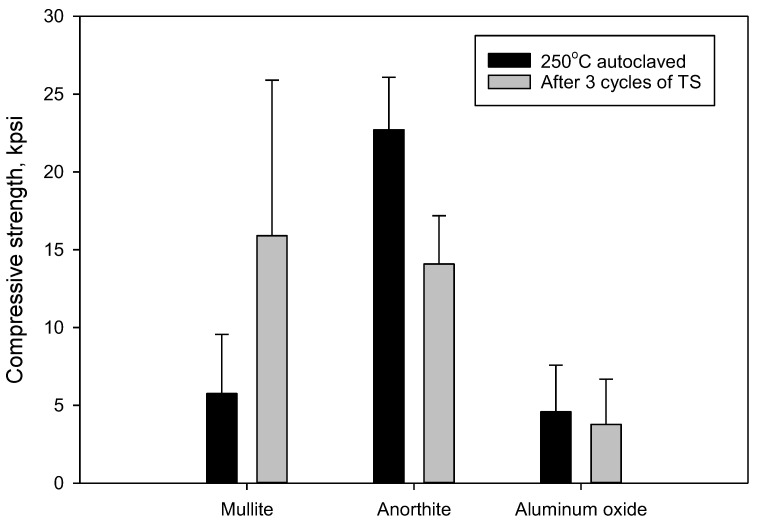
Compressive strength of 250 °C autoclaved samples and samples exposed to 3 cycles of **thermal shock** (one cycle: 600 °C heating overnight → cold water quenching for 10 min).

**Figure 24 materials-15-06703-f024:**
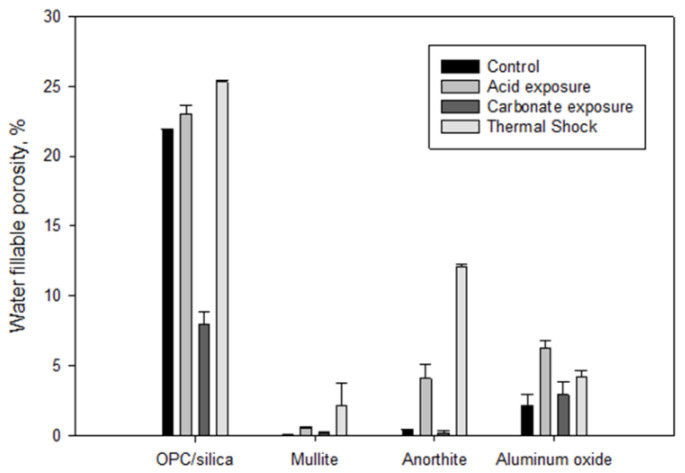
Water fillable porosity of different materials before and after imposed stress conditions. “Control” refers to the 250 °C autoclaved samples.

**Figure 25 materials-15-06703-f025:**
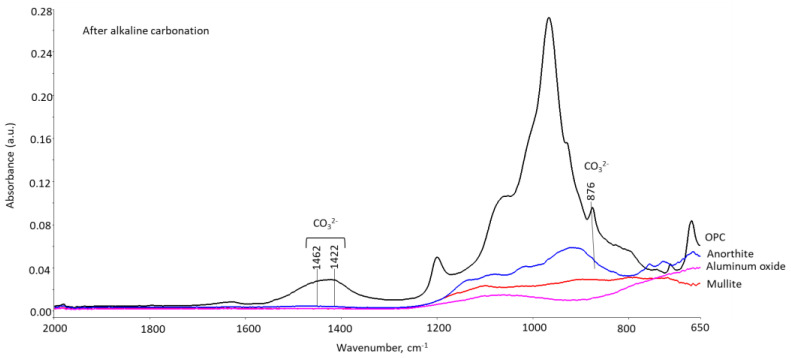
FTIR spectra of tested materials after exposure to alkaline carbonate (250 °C, 40 days) and 3 cycles of thermal shock (one cycle: 600 °C heat overnight → cold water quenching for 10 min).

**Figure 26 materials-15-06703-f026:**
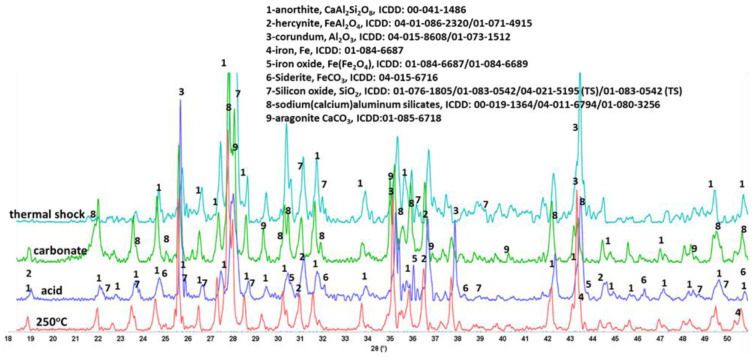
XRD patterns of **anorthite** samples after the autoclaving at 250 °C, acid-, alkaline carbonate-, and thermal shock tests.

**Figure 27 materials-15-06703-f027:**
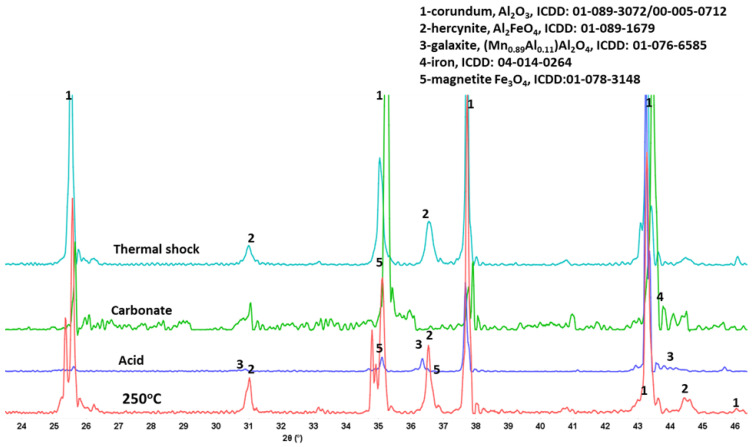
XRD patterns of **aluminum oxide** samples after the autoclaving at 250 °C, acid-, alkaline carbonate-, and thermal shock tests.

**Figure 28 materials-15-06703-f028:**
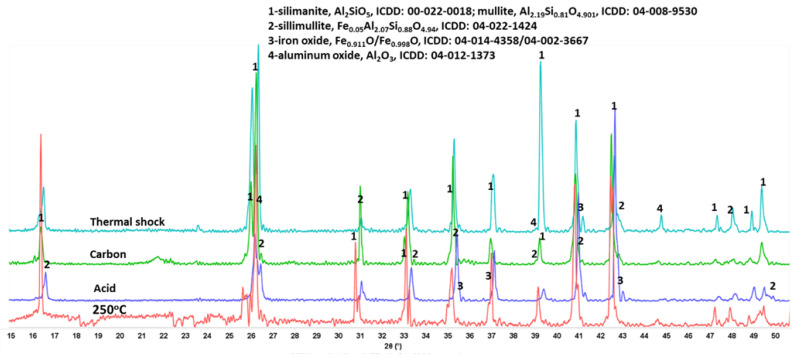
XRD patterns of **mullite** samples after the autoclaving at 250 °C, acid-, alkaline carbonate-, and thermal shock tests.

**Figure 29 materials-15-06703-f029:**
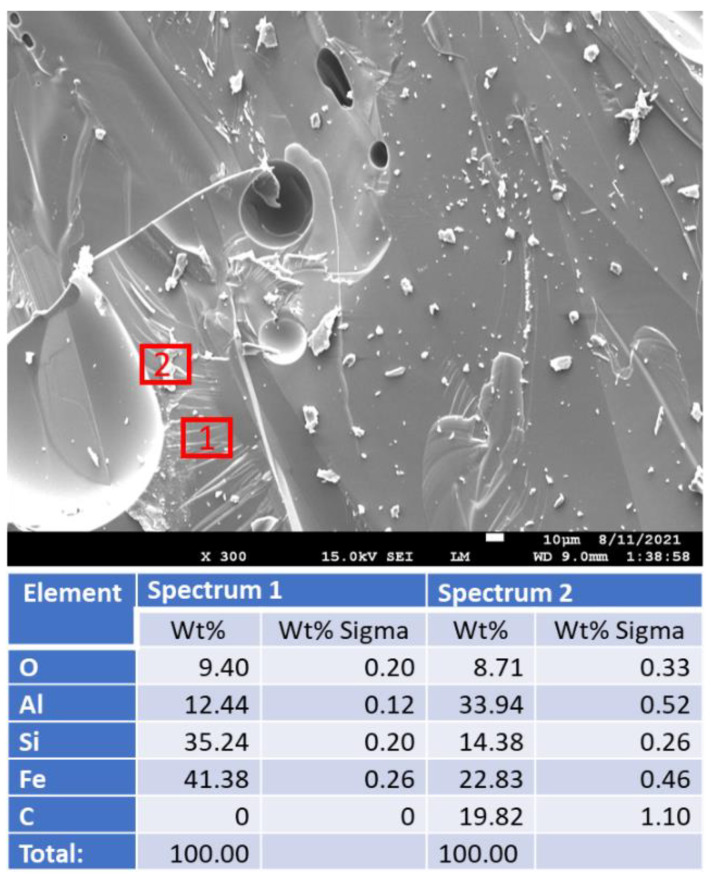
Photomicrograph of **mullite** sample with the elemental composition of typical sites shown by red boxes after alkaline carbonation at 250 °C for 40 days.

**Figure 30 materials-15-06703-f030:**
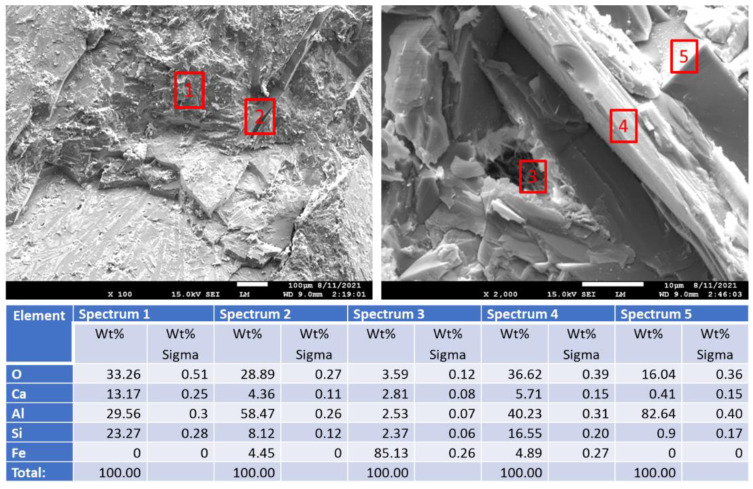
Photomicrographs of **anorthite** samples with the elemental composition of typical sites shown by red boxes after alkaline carbonation at 250 °C for 40 days.

**Figure 31 materials-15-06703-f031:**
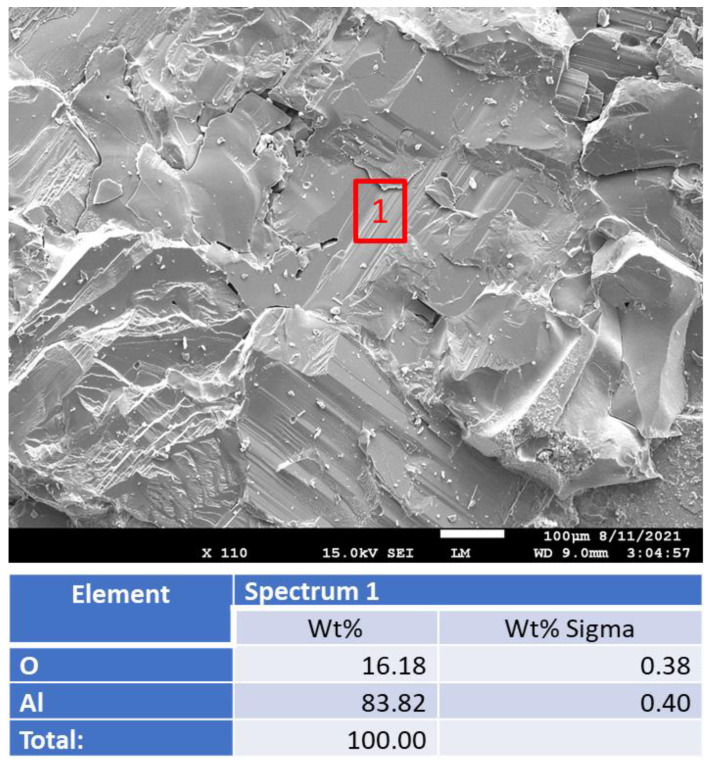
Photomicrograph of an **aluminum-oxide** sample with the elemental composition of a typical site shown by the red box after alkaline carbonation at 250 °C for 40 days.

**Figure 32 materials-15-06703-f032:**
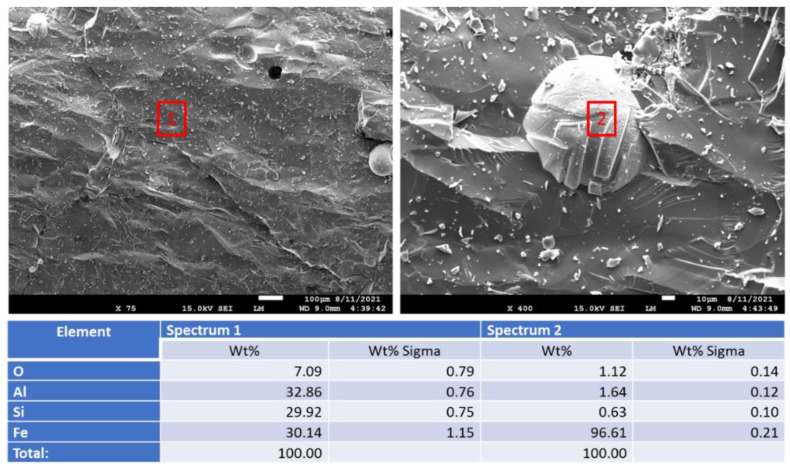
Photomicrographs of **mullite** samples with the elemental composition of typical sites shown by the red boxes after 3 cycles of thermal shock (one cycle: 600 °C heat → cold water quenching).

**Figure 33 materials-15-06703-f033:**
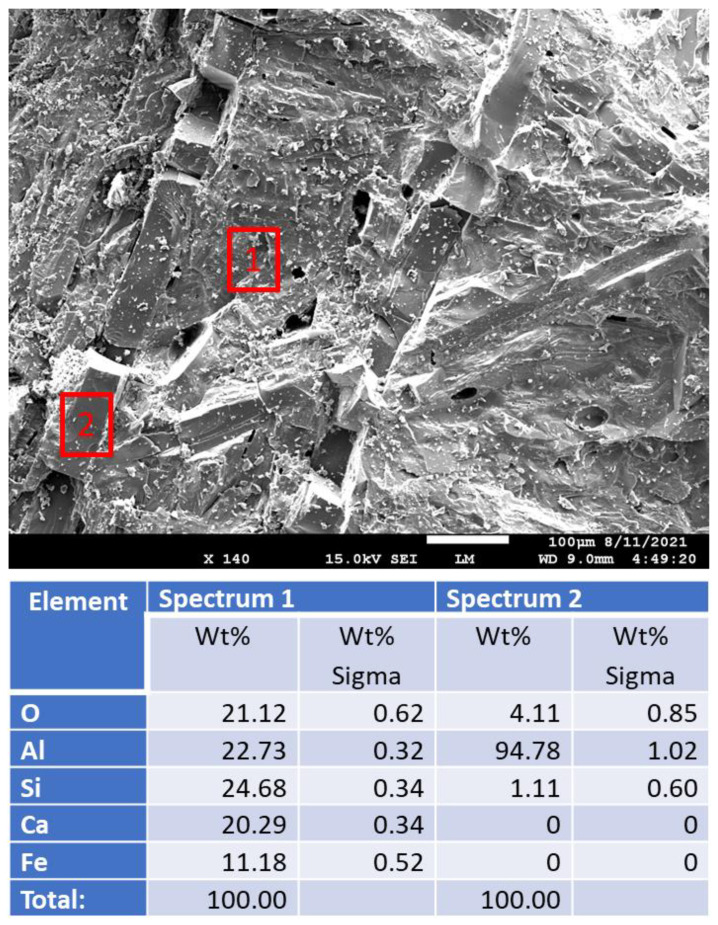
Photomicrograph of an **anorthite** sample with the elemental composition of typical sites shown by the red boxes after 3 cycles of thermal shock (one cycle: 600 °C heat → cold water quenching).

**Figure 34 materials-15-06703-f034:**
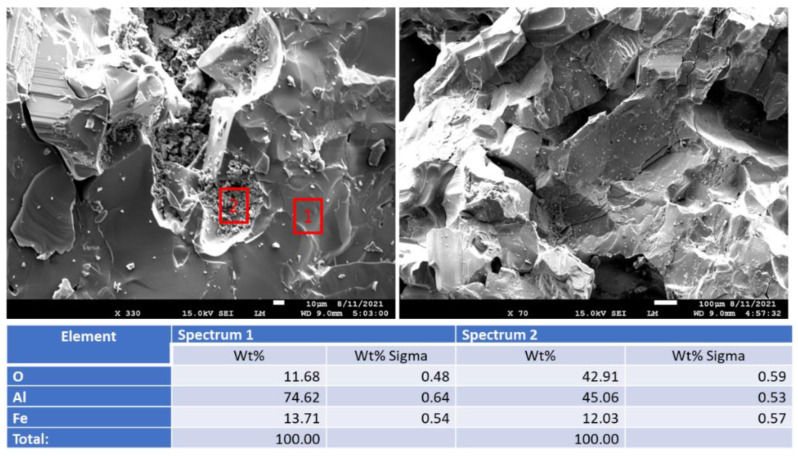
Photomicrographs of **aluminum oxide** samples with the elemental composition of typical sites shown by the red boxes after 3 cycles of thermal shock (one cycle: 600 °C heat → cold water quenching).

**Table 1 materials-15-06703-t001:** Elemental and crystalline compositions of the samples before exposure tests.

Sample Name	Averaged Elemental Composition, Major (Minor), (Expressed in the Form of Oxides, Mass %)	Crystalline Composition (ICDD Compound Number)
Aluminum oxide	Al_2_O_3_—60%, Fe_2_O_3_—5% (major) CaO—0.3%, MnO—0.2% (minor)	corundum, Al_2_O_3_, (01-089-3072/00-005-0712) (major) hercynite, Al_2_FeO_4_, (01-089-1679) galaxite, (Mn_0.89_Al_0.11_) Al_2_O_4_, (01-076-6585) iron, (04-014-0264) magnetite Fe_3_O_4_, (01-078-3148)
Anorthite	Al_2_O_3_—45%, SiO_2_—27%, CaO—15%, Fe_2_O_3_—10% (major) Cl—0.7% (minor)	anorthite, CaAl_2_Si_2_O_8_, (00-041-1486) (major) hercynite, FeAl_2_O_4_, (04-01-086-2320/01-071-4915) corundum, Al_2_O_3_, (04-015-8608/01-073-1512) iron, Fe, (01-084-6687) iron oxide, Fe(Fe_2_O_4_), (01-084-6687/01-084-6689)
Aluminum-silicate	Al_2_O_3_—60%, SiO_2_—30%, Fe_2_O_3_—10% (major) MnO—0.4%, CaO—0.4%, Cr_2_O_3_—0.1% (minor)	sillimanite, Al_2_SiO_5_, (00-022-0018); mullite, Al_2.19_Si_0.81_O_4.901_, (04-008-9530) sillimullite, Fe_0.05_Al_2.07_Si_0.88_O_4.94_, (04-022-1424) iron oxide, Fe_0.911_O/Fe_0.998_O, (04-014-4358/04-002-3667)

## Data Availability

The data presented in this study are available on request from the corresponding author.

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
