# Peer review of "Corrosion-Resistant Plug Materials for Geothermal Well Fluid Control"

_materials, 2022, doi:10.3390/ma15196703_

Round 1

Reviewer 1 Report

The reviewed work presents detailed studies of new materials used in aggressive geothermal environments. The research results are very extensive and the conclusions are fully convincing. I recommend the work for publication in its current form.

Author Response

We thank the reviewer for such a positive evaluation of our work.

Reviewer 2 Report

This paper has carried out a systematic study on Plug and abandonment. A range of engineered thermite systems which yield thermite reaction products was developed. These materials were evaluated for strong acid resistance, carbonate resistance and thermal shock resistance. This work are sufficient and has engineering significance. The following suggestions may need to be considered before publication:

1. Introduction: the current research progress of thermite plugs and platforms is not well described. So, the innovation points of this paper are not clear enough.

2. The picture definition needs to be improved, especially Fig. 5 (reflecting the corrosion phenomenon).  

3. Figs.18-20: As we all know, SEM observation is a glimpse of the truth. As shown in Figures, the size of SEM photomicrographs is 10-100um, far smaller than the research object. So, the Ca content from one point in SEM photomicrographs can't reflect the overall corrosion situation. It is recommended to take values of 2-3 points.

4. English needs further improvement. The phenomenon, that one sentence is one paragraph, occurs many times in this paper.

Reviewer 3 Report

The paper entitled "Corrosion-resistant Plug Materials for Geothermal Well Fluid Control" is insightful and well-written. The authors carried out series of tests to understand the durability of materials. However, there is some missing information related to methodology of the research and reasons of selection of conditions (exposure temperature and concentration of acids, etc.) There is also a room from improving the technical discussion and supporting/correlating with the findings in the literature. The manuscript can be accepted after addressing the minor comments annotated in the attached pdf file.

Author Response

Thank you for your comments.

  1. We did not use an ASTM method for compression tests of the samples. The materials are not intended for building and construction use, they are designed for the use in underground wells. In the case of underground wells ASTM standards are not used for testing mechanical properties of cementitious materials. The use of Instron mechanical testing equipment is widely accepted both in oil&gas industry and in geothermal wells materials testing for evaluation of cement samples properties. Since in our evaluations we compared performance of thermite samples against that of commonly used high temperature Portland cement formulation and since both reference and exposed samples were tested the results of testing under identical conditions were used to make conclusions on  materials performance.
  2. We modified the chapter on the appearance of the samples, adding some discussion. However, for the most part the discussion of the samples acid corrosion is in a separate section "discussion" included after all the experimental evidence are presented to make more informed analysis. The references of the papers that you provided came cut off in the attached pdf. version of the manuscript so unfortunately we could not use them.
  3. Indeed the XRD results presented in Figure 15 do not show any gypsite. This is in agreement with the visual observation of mullite samples that showed only minor samples' erosion. That means that whatever corrosion products formed, they stayed dissolved without any significant precipitations on the surface. The slight decrease in samples' crystallinity reported in the visual observations also agrees with the lack of crystalline  corrosion products on their surfaces. We added some relevant notes to the text of the manuscript.